# Polyethylene Glycol and Polysorbate 80 Skin Tests in the Context of an Allergic Risk Assessment for Hypersensitivity Reactions to Anti-SARS-CoV-2 mRNA Vaccines

**DOI:** 10.3390/vaccines11050915

**Published:** 2023-04-28

**Authors:** Emanuele Nappi, Francesca Racca, Alessandra Piona, Maria Rita Messina, Sebastian Ferri, Donatella Lamacchia, Giuseppe Cataldo, Giovanni Costanzo, Lorenzo Del Moro, Francesca Puggioni, Giorgio Walter Canonica, Enrico Heffler, Giovanni Paoletti

**Affiliations:** 1Personalized Medicine, Asthma and Allergy, IRCCS Humanitas Research Hospital, 20089 Rozzano, Italy; emanuele.nappi@humanitas.it (E.N.); francesca.racca@humanitas.it (F.R.); mariarita.messina@humanitas.it (M.R.M.); sebastian.ferri@humanitas.it (S.F.); donatella.lamacchia@humanitas.it (D.L.); giuseppe.cataldo@humanitas.it (G.C.); giovanni.costanzo@humanitas.it (G.C.);; 2Allergy Service, Humanitas San Pio X Hospital, 20159 Milano, Italy; alessandra.piona@sanpiox.humanitas.it; 3Department of Clinical and Experimental Medicine, University of Florence, 50121 Firenze, Italy; 4Department of Biomedical Sciences, Humanitas University, 20072 Pieve Emanuele, Italy

**Keywords:** SARS-CoV-2, vaccine, allergy, polyethylene glycol, polysorbate, skin tests

## Abstract

Concern has arisen about hypersensitivity reactions in patients with allergic reactions to drugs containing polyethylene glycol (PEG) or polysorbate 80 (PS80), excipients of currently available anti-SARS-CoV-2 mRNA vaccines. However, the actual utility of PEG and PS80 skin allergy testing is currently still debated. We retrospectively analyzed all cases of patients on whom we performed allergometric skin tests for PEG and PS80 in the context of a pre-vaccination screening (for patients with multiple hypersensitivity reactions to drugs for which these excipients were among the suspected agents) or following suspected hypersensitivity reactions to anti-SARS-CoV-2 vaccines. A total of 134 tests were performed for PEG and PS80, eight of which produced uninterpretable results (due to dermographism or non-specific reactions). Of the remaining 126 cases (85 pre-vaccinal and 41 post-vaccine reactions), 16 (12.7%) were positive for PEG and/or PS80. Stratifying by clinical indication, there were no statistically significant differences in the proportion of positive tests between patients evaluated in the context of the pre-vaccination screening and those evaluated after a vaccine reaction (10.6% vs. 17.1%, respectively, *p* = 0.306). Allergometric skin tests for PEG and PS80 in our case series resulted positive in an unexpectedly high proportion of patients, suggesting that testing for allergy to these two excipients should not be ignored in case of reasonable clinical suspicion.

## 1. Introduction

Since the very beginning of the vaccination campaign, an unexpectedly high number of anti-SARS-CoV-2 vaccine-related anaphylaxis cases have been observed, with official reports suggesting a rate of 2.5 to 11.1 reactions per million doses depending on the specific vaccine [1,2]. These figures suggest that anti-SARS-CoV-2 vaccine allergies are still very rare and should not pose a substantial limit to the progress of the vaccination campaign. Nevertheless, studies antecedent to the COVID-19 pandemic report rates of vaccine anaphylaxis approximating one case per one million administered doses [3], which is lower than the one reported for anti-SARS-CoV-2 vaccines. Most anaphylactic reactions to anti-SARS-CoV-2 vaccines have been reported in females (from 90 to 100%), usually with a history of previous anaphylaxis or allergy (reported in 81–90% of cases) [1,2]. Reactions occurred within a range of 2 to 150 min following vaccine administration, with the vast majority occurring within the first 15 min [2]. These reports include only anaphylactic reactions, as classified by the Brighton Collaboration case definition (only cases with a high degree of diagnostic certainty were included, namely Brighton levels 1 and 2) [4]. An additional number of non-anaphylactic hypersensitivity reactions occurred as well [1,2].

Given the utmost importance of anti-SARS-CoV-2 vaccines in hindering the progress of the pandemic, intense efforts to understand the mechanisms behind such reactions have been made. Polyethylene glycol (PEG) is one of the few excipients of mRNA-based anti-SARS-CoV-2 vaccines (e.g., mRNA-1273 and BNT162b2) with allergenic potential [5]. The mRNA molecules contained in these vaccines are stabilized by lipid nanoparticles, which are eventually PEGylated to improve delivery to target cells [6]. PEG is widely applied in pharmaceutical, cosmetic, edible, and household products [7]. It is obtained from the polymerization of ethylene glycol and can be found at different molecular weights (MW) depending on the chain length. Although the chemical structure of different PEGs (i.e., at different MWs) is identical, a higher MW is more likely to cause allergy, possibly due to the higher number of epitope units available for IgE binding [7,8]. The PEG contained in mRNA-based anti-SARS-CoV-2 vaccines has an MW of 2000 kDa. Studies of hypersensitivity reactions following mRNA-based anti-SARS-CoV-2 vaccination found PEG to be a suspect culprit molecule, as proven via positive allergy tests and compatible past history [7]. Patients with a PEG allergy typically report multiple systemic reactions to unrelated drugs and possibly also numerous systemic or localized allergies to topical cosmetic products [7,9].

Polysorbate 80 (PS80) has been proposed to mediate viral vector-based anti-SARS-CoV-2 vaccines (e.g., AZD1222 and Ad26.COV2.S) [10]. Polysorbates (PS) are composed of a sorbitan molecule linked to four short PEG chains, one of which is bound with a fatty acid. The number that follows PS depends on the type of fatty acid associated with it (monooleate for PS80). PSs contribute to the solubility of the anti-SARS-CoV-2 vaccine and are also contained in other vaccines (e.g., hepatitis B, influenza, and papilloma virus vaccines). PS80 is widely applied in pharmaceutical (e.g., depot steroids) and food preparations; moreover, it is frequently applied as an emulsifier in essential oils. Recently, increased attention to PS allergy has arisen, but the role of PSs specifically in the context of vaccine allergies had not been emphasized before the introduction of anti-SARS-CoV-2 vaccines and still needs to be defined [11].

Other anti-SARS-CoV-2 vaccine excipients have been proposed as potential allergens, but their link with hypersensitivity reactions remains unclear to some extent. Tromethamine is contained in the mRNA-1273 vaccine; its potential role in allergies was proposed following the observation of cross-reactive allergy to iodine- and gadolinium-based contrast media preparations that contained only tromethamine as a common molecular component and eventually confirmed using positive cutaneous allergy tests [12]. Disodium edetate dihydrate (EDTA), contained in the AZD1222 vaccine, was also linked to allergic reactions in an individual receiving local anesthetics and contrast media formulations containing it [13]. Additionally, phospholipids and nanoparticles contained in anti-SARS-CoV-2 vaccines may play a role in some hypersensitivity cases, probably through IgE-independent mechanisms; however, their role in this context still needs to be elucidated [14].

IgE-mediated mast cell degranulation through activation of the Fcε receptor-1 is a well-known mechanism that might underlie immediate hypersensitivity reactions to vaccines. There are also other, less understood pathways through which mast cell degranulation can occur and which have been linked to vaccine hypersensitivity. For example, mast cells might be directly activated by mediators generated via complement activation, called anaphylatoxins, or through stimulation of the Mas-related G-protein-coupled receptor X2 (MRGPRX2) [15]. IgE-independent mechanisms seem to be involved in some anti-SARS-CoV-2 hypersensitivity reactions and deserve further investigation [15,16].

The actual utility of PEG and PS80 skin allergy testing for identifying patients at risk of allergic reactions to SARS-CoV-2 vaccines and the role of these excipients in the development of allergic reactions to vaccines are currently still debated. We hereby describe our case series of patients who were subjected to PEG and PS80 skin allergy testing in the context of the evaluation of allergic risk to anti SARS-CoV-2 vaccines.

## 2. Materials and Methods

We retrospectively analyzed all cases of patients who underwent allergometric skin tests (skin prick tests—SPTs and intradermal tests—IDTs) for PEG and PS80. Patients underwent testing either following a history of multiple allergic reactions to chemically unrelated drugs containing PEG and/or PS80 (pre-vaccine evaluation group) or following suspected hypersensitivity reactions to anti-SARS-CoV-2 vaccines (post-vaccine reaction group) in the period between May 2021 and November 2022. Demographic and clinical data were collected. In detail, patients in the pre-vaccine evaluation group received indication for testing if they reported immediate (occurring within one hour of consumption) hypersensitivity reactions to at least two chemically unrelated drugs containing PEG or PS80 and if no other drugs containing PEG (at MW > 2000 kDa) and PS80 were consumed and tolerated in the past. Patients in the post-vaccine reaction group were selected for testing if they reported an immediate hypersensitivity reaction following any dose of an anti-SARS-CoV-2 vaccine. In the post-vaccine reaction group, testing was performed within 2–8 weeks from the suspected hypersensitivity reaction. In contrast, the time passed between the last hypersensitivity reaction and testing was not uniform in the pre-vaccinal evaluation group, and we did not apply time limits in this group.

Allergy testing was performed by trained allergists working at our institution. The procedure of allergometric tests is reported in Table 1. A modified version of the methodology proposed by Banerji et al. was applied [17]. Prick tests were performed using a 100 mg/mL preparation for colonoscopy (Normalene^®^) to test for PEG 3350 and pure PS80 to test for PS80. Concentrations of 40 mg/mL intradermal methylprednisolone prednisolone acetate (Depo-Medrol^®^) and 40 mg/mL triamcinolone acetonide (Kenacort^®^) were used to test for test for PEG 3350 and PS80, respectively. The only diluent used was physiologic saline (NaCl 0.9%). Patients were asked to avoid the use of antihistamines and corticosteroids in the two weeks preceding the test. Patients who tested positive to prick tests were not subjected to intradermal testing. Five healthy controls, without history of allergic diseases, were recruited from our unit to assess the non-irritating effect of the solutions used for allergy testing; none of them had a positive test or skin reaction to any of the skin pricks or intradermal PEG- or PS80-containing solutions.

Statistical analysis was performed using SPSS 20.0 software (SPSS, Chicago, IL, USA). The Kolmogorov–Smirnov test was used to evaluate the normality of distribution of each continuous variable, and depending on the result of this test, the Student *t*-test or Mann–Whitney test was used to compare continuous variables. Categorical variables were compared using Fisher’s exact test. Continuous variables were presented as mean ± standard deviation (SD). Finally, *p* values of <0.05 were considered statistically significant.

All patients signed an informed consent agreement for the desensitization procedure and for publication of anonymized data.

## 3. Results

A total of 134 allergometric skin tests for PEG and PS80 were performed. The characteristics of our patients and test results are summarized in Table 2. Eighty-five patients (63.4%) underwent testing due to a history of multiple allergic reactions to chemically unrelated drugs containing PEG and/or PS80 (pre-vaccinal evaluation group), and forty-one (30.6%) underwent testing due to a suspected hypersensitivity reaction following an anti-SARS-CoV-2 vaccine (post-vaccine reaction group). Eight patients (6.0%) had uninterpretable test results (due to dermographism or non-specific reactions) and were therefore excluded from the subsequent analysis. Among the remaining 126 patients with interpretable skin tests results, we observed 16 (12.7%) positive test results (Table 2). The proportion of positive tests was 10.6% in patients evaluated in the context of pre-vaccination screening, and 17.6% in those evaluated after a vaccine reaction, without a statistically significant difference between the two groups *(p =* 0.306) (Table 2). Among the 41 patients in the post-vaccine reaction group, 30 experienced an immediate hypersensitivity reaction following the administration of the BNT162b2 vaccine, 8 experienced one following the mRNA1273 vaccine, and 3 experienced one following the AZD1222 vaccine.

Interestingly, 9 out of 16 patients with positive skin tests (56.2%) were positive to both PEG and PS80, accounting for 66.7% and 42.9% of those evaluated in the context of a pre-vaccination screening and those evaluated after a vaccine reaction, respectively (*p* = 0.341). Among the 11 patients who tested positive for PEG, 4 had positive prick tests and the remaining 7 had positive intradermal tests. Among the 14 patients who tested positive for PS80, 6 had positive prick tests and the remaining 8 had positive intradermal tests.

Four patients (25.0% of patients who tested positive via skin tests) developed an anaphylactic reaction during the skin tests. None of the patients experiencing reactions during testing required hospitalization and all fully recovered.

## 4. Discussion

PEG and structurally related PS80 have been indicated as culprit allergens related to some anti-SARS-CoV-2 vaccine hypersensitivity reactions, but the relevance of PEG and PS80 allergies in this context still needs to be fully elucidated [7,13]. There has been a vivid collaboration between our Allergy Unit and Vaccination Center to make vaccine delivery as safe as possible in individuals with a high-risk profile [18]. Patients underwent testing either due to hypersensitivity reactions following an anti-SARS-CoV-2 vaccine or due to a history of multiple allergic reactions to structurally unrelated drugs containing PEG and/or PS80. We performed 134 allergometric skin tests for PEG and PS80, obtaining 126 interpretable results. The remaining eight were excluded due to uninterpretable test results (dermographism or non-specific reactions). Notably, after an appropriate clinical and diagnostic evaluation, safe vaccine delivery should be possible for most patients, and only a minority are deemed ineligible to vaccination; the latter should wait for the release of new vaccines with distinct allergenic profiles [19].

PEG and PS allergies have been reported to be rare [9,10]. Allergometric skin tests for PEG and PS80 in our case series resulted positive in 12.7% of cases, a figure that is unexpectedly high. It is true that our patients were highly selected, with a relatively high pre-test probability, but still, this result indicates that allergy to PEG and its derivatives must be considered in patients with a history of allergy to multiple unrelated drugs. Moreover, PEG and PS80 allergometric skin testing seems to be useful in the evaluation of patients at high risk of hypersensitivity reactions to anti-SARS-CoV-2 vaccines; nevertheless, the predictive capacity of PEG and PS80 testing in this context still needs to be defined, as not all patients who test positive eventually experience a reaction upon vaccination [20]. The rate of positive tests was slightly higher in the patients undergoing testing after post-vaccine reactions than in patients with a suggestive history of PEG and/or PS80 hypersensitivity (17.1% vs. 10.6%). A similar study by Montera et al. [21] reported lower rates of positive allergometric skin tests to PEG and PS80 in patients deemed at risk of anti-SARS-CoV-2 vaccine hypersensitivity, specifically 10/362 (2.8%) in patients with a history compatible with PEG and/or PS80 drug allergy and 12/169 (7.1%) in patients experiencing immediate hypersensitivity reactions to anti-SARS-CoV-2 vaccines. As in our study, they noted a lower positivity rate in the pre-vaccinal evaluation group than in the post-vaccine reaction group (10.6% vs. 17.1%, respectively, in our study). The clinical implications of our findings deserve further investigation, and more studies are necessary to assess whether patients with positive allergometric tests should actually be considered ineligible for vaccination.

Notably, 7% of our patients (56% of those with positive skin tests) reacted positively for both PEG and PS80 allergometric skin tests. Although we were able to prove cutaneous cross-reactivity, the clinical history of some of our patients was compatible with cross-reactive allergy to both components. The molecular similarity between PEG and PS80 would explain why cross-reactivity has been observed in our patient sample, and there are studies reporting cross-reactive allergies to PEG and PS [22,23,24].

Moreover, 25% of patients with positive skin tests for PEG and/or PS80 developed systemic reactions during the testing session. Systemic reactions to PEG allergometric tests have been reported to occur [9], suggesting that a high level of caution should be adopted when performing these tests. It is possible that intradermal tests increase the risk of experiencing such reactions [9]. Nevertheless, it is reported that some PEG-allergic patients have negative SPTs and positive IDTs [25], suggesting IDTs should be considered when SPTs are negative and occur in the absence of signs of systemic hypersensitivity [26]. More research in this field is required to better assess the best diagnostic pathway for patients with suspected allergies to PEG and PSs.

Among patients who underwent testing for hypersensitivity reactions to previous doses of anti-SARS-CoV-2 vaccines, 32 (78%) tested negative. It is possible that reactions in these cases were mediated by other excipients, as other anti-SARS-CoV-2 vaccine components have been proposed to have the capability to mediate hypersensitivity reactions (e.g., tromethamine, EDTA); however, their role still needs to be defined [13]. Moreover, some reactions which occurred following anti-SARS-CoV-2 vaccines were proven to be independent from the allergic mechanism (no serum tryptase elevation, no repeatability) [27]. While there is evidence that PEG hypersensitivity can be IgE-mediated [23], the potential role of IgEs in PS80 hypersensitivity reactions is still unclear, and IgE-independent mechanisms have been proposed to mediate both PEG and PS allergies [28]. Given PEG’s molecular structure, it is reasonable to suppose that PEG allergy is mediated by one potential epitope (repeated multiple times) [26]; however, its immunogenic profile might be modified when processed for specific preparations, for example, when linked to mRNA vaccine nanoparticles [29]. In contrast, PSs have more potential epitopes for allergy, one of which is fully shared with PEG, and some of which are not. A study showed significantly higher levels of PEG-specific IgEs but not PS80-specific IgEs in patients who experienced allergic reactions following the administration of anti-SARS-CoV-2 vaccines compared to non-allergic controls [25]. Even if PS80-specific IgEs were not significantly elevated, on average, they were higher compared to controls. This brings about the idea that PS80-specific IgE can still be involved in anti-SARS-CoV-2 vaccine hypersensitivity, but to a lesser extent than PEG-specific IgEs. PEG-specific IgEs were detected some cases, providing insights into IgE-independent mechanisms also in the context of PEG hypersensitivity [25]. To conclude, the mechanisms of PEG and PS80 hypersensitivity are not yet fully understood, and we were not able to properly assess this aspect in this study. More research on this topic, potentially combining multiple test modalities that also assess IgE-independent pathways, is warranted.

Other anti-SARS-CoV-2 vaccine excipients have been linked to hypersensitivity reactions, including trometamol, EDTA, phospholipids, and nanoparticles [12,13,14]. Unfortunately, we were not able to define the relevance of excipients other than PEG and PS80 in this study, so further research on this topic is needed. Another limit of our study is that we were not able to perform allergometric skin tests with original vaccine extracts as this research was conducted during an emergency period of mass vaccination, so they were not available.

## 5. Conclusions

In conclusion, we observed a surprisingly high number of positive allergometric tests to PEG and PS80 in the context of a pre-vaccinal evaluation. Our results suggest that PEG and PS allergies should be considered in patients reporting a history of hypersensitivity reactions to anti-SARS-CoV-2 vaccines and/or of multiple hypersensitivity reactions to chemically unrelated drugs. The mechanisms behind PEG and PS80 hypersensitivity seem to extend beyond type 1 hypersensitivity. More research on PEG and PS80 hypersensitivity mechanisms and diagnosis is needed.

## Figures and Tables

**Table 1 vaccines-11-00915-t001:** Procedure of allergometric skin tests.

Step		Tested Drug	Dilution	Cumulative Time (min)
1	Positive control	Histamine	1:1	0
	Negative control	Glycerin	1:1	
	Prick test	PEG 3350	1:100	
	Prick test	Polysorbate 80	1:100	
2	Prick test	PEG 3350	1:10	30
	Prick test	Polysorbate 80	1:10	
3	Prick test	PEG 3350	1:1	60
4	Intradermal	Methylprednisolone acetate (Depo-Medrol) 40 mg/mL	1:100	90
	Intradermal	Triamcinolone acetonide (Kenacort) 40 mg/mL	1:100	
5	Intradermal	Methyl-prednisolone Acetate (Depo-Medrol) 40 mg/mL	1:10	120
	Intradermal	Triamcinolone acetonide (Kenacort) 40 mg/mL	1:10	

**Table 2 vaccines-11-00915-t002:** Demographic characteristics and allergometric tests results.

	Total	Pre-Vaccinal Evaluation Group	Post-Vaccine Reaction Group	*p*-Value
Number	126	85	41	
Age, years ± SD	50.2 ± 16.7	51.7 ± 17.1	46.9 ± 15.6	0.101
Sex, num. females (%)	102 (81.0%)	66 (77.6%)	36 (87.8%)	0.174
Positive allergometric test N (proportion in %)	16 (12.7%)	9 (10.6%)	7 (17.6%)	0.306
PEG	11 (8.7%)	7 (8.2%)	4 (9.8%)	0.777
PS80	14 (11.1%)	8 (9.4%)	6 (14.6%)	0.382
PEG plus PS80	9 (7.1%)	6 (7.1%)	3 (7.3%)	0.356

## Data Availability

Not applicable.

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
