# Peer review of "Polyethylene Glycol and Polysorbate 80 Skin Tests in the Context of an Allergic Risk Assessment for Hypersensitivity Reactions to Anti-SARS-CoV-2 mRNA Vaccines"

_vaccines, 2023, doi:10.3390/vaccines11050915_

Round 1
Reviewer 1 Report
This brief report retrospectively analyzed all the cases of patients to whom they performed allergometric skin tests for PEG and PS80 in the context of a pre-vaccination screening or to following suspected hypersensitivity reactions to anti-SARS-CoV-2 vaccines and found an unexpectedly high proportion of positive patients.
I have some comments.
1. As the authors presented in the introduction, PEG is one of the few excipients of mRNA-based anti-SARS-CoV-2 vaccines, while PS80 has been proposed to mediate viral vector-based anti-SARS-CoV-2 vaccines. These two excipients are used in different types of vaccines, thus the risk of allergic reactions could be varying in different types of anti-SARS-CoV-2 vaccines. The authors should ensure that all the subjects were gave the same type of anti-SARS-CoV-2 vaccine.
2. What about the contribution to allergic risk of other components in the investgated anti-SARS-CoV-2 vaccine?
3. Whether the time for allergometric skin tests uniform after vaccination?
4. This report did not fully elucidate the mechanism of PEG and PS80 hypersensitivity. It would be better to show the IgE and IgG levels in the subjects.
Author Response
Dear Reviewer,
thank you very much for your time and your valuable feedback. We modified our manuscript according to your suggestions as follows (point-by-point response to your comments):
- As the authors presented in the introduction, PEG is one of the few excipients of mRNA-based anti-SARS-CoV-2 vaccines, while PS80 has been proposed to mediate viral vector-based anti-SARS-CoV-2 vaccines. These two excipients are used in different types of vaccines, thus the risk of allergic reactions could be varying in different types of anti-SARS-CoV-2 vaccines. The authors should ensure that all the subjects were gave the same type of anti-SARS-CoV-2 vaccine.
RESPONSE: We included details about the type of vaccine that caused hypersensitivity reactions in the post-vaccine reaction group (rows 158-161)
2. What about the contribution to allergic risk of other components in the investgated anti-SARS-CoV-2 vaccine?
RESPONSE: One of the limits of our study is that we tested only for PS80 and PEG, and not for other anti-SARS-CoV-2 vaccine excipients which could be potentially involved in some of the hypersensitivity reactions in the post-vaccine reaction group. The potential role of other vaccine excipients is presented in the introduction (rows 79-89) and we specified in our discussion that the limits of our study include that we did not assess the relevance of other excipients (rows 264-267).
3. Whether the time for allergometric skin tests uniform after vaccination?
RESPONSE: We specified that in the post-vaccine reaction group, allergy testing was performed within 2 and 8 weeks from the hypersensitivity reaction. Instead, we did not apply any limits in terms of time passed from the last hypersensitivity reaction in the pre-vaccinal evaluation group (rows 118-121)
4. This report did not fully elucidate the mechanism of PEG and PS80 hypersensitivity. It would be better to show the IgE and IgG levels in the subjects.
RESPONSE: We do agree with you in that PEG and PS80 hypersensitivity mechanisms are not yet fully understood and that it would be relevant to better assess this aspect. Unfortunately, we did not include in our study the measurement of PEG- and/or PS80-directed antibodies. We do realize that one of the limits of this research is that we did not extensively investigate the mechanisms of PEG and PS80 hypersensitivity and we specified it in the discussion as a limit of our study (rows 260-263)
Reviewer 2 Report
Dear Authors,
you have provided important results. Nevetheless, the manuscript should be improved.
Please, provide:
- extensive description of the dilutions for allergen substances and diluents;
- prooves for un-irritable concentrations for chemical and drug substances;
- all data of allergen skin tests for each allergen tested, differentiate PEG, PS, and steroid preparations;
- all criteria for pre-vaccination group (types of reaction, culpit drugs etc) as well as for post-vaccination group.
Author Response
Dear Reviewer,
thank you very much for your time and your valuable feedback. We modified our manuscript according to your suggestions as follows (point-by-point response to your comments):
- extensive description of the dilutions for allergen substances and diluents;
RESPONSE: We explained more clearly the components of allergen skin testing and we specified the diluent used (rows 123-128)
- prooves for un-irritable concentrations for chemical and drug substances;
RESPONSE: We included 5 healthy controls (without history of allergic diseases) to assess the non-irritating effect of the solutions used for allergy testing; none of them had a positive test or skin reaction to any of the skin prick or intradermal PEG or PS80 solutions (rows 130-133).
- all data of allergen skin tests for each allergen tested, differentiate PEG, PS, and steroid preparations
RESPONSE: We added in our results (rows 165-167) details about each allergen tested, differentiating prick tests and intradermal tests.
- all criteria for pre-vaccination group (types of reaction, culpit drugs etc) as well as for post-vaccination group.
RESPONSE: We described more extensively the selection criteria for allergy tests of both groups (rows 112-118).
Reviewer 3 Report
The frequency of skin test positive responses to PEG and PS80 reported here are indeed very high even for a hypersensitivity-prone sample. There are few similar studies reported with most reports being case studies. In comparing the frequency of responses with the frequency found for COVID vaccination the clinical conclusion drawn from the results might better be the SPT positivity is not very useful. This should be better discussed.
A similar report has just been published by Montera et al. "The role of skin tests with polyethylene glycol and polysorbate 80 in the vaccination campaign for COVID-19: results from an Italian multicenter survey." (doi: 10.23822/EurAnnACI.1764-1489.291.). The results here should be critically compared to those in this report. In doing so it should be noted that Montera et al. have trouble with their arithmetic in that 10/362 is 2% not 0.02% and most people would not consider that rare.
Given the high frequency of positive results the testing methodology and controls and what is considered positive becomes very important. The wide-ranging review of reference 15 does not give the information required. More precise methods are needed.
Author Response
Dear Reviewer,
thank you very much for your time and your valuable feedback. We modified our manuscript according to your suggestions as follows (point-by-point response to your comments):
- The frequency of skin test positive responses to PEG and PS80 reported here are indeed very high even for a hypersensitivity-prone sample. There are few similar studies reported with most reports being case studies. In comparing the frequency of responses with the frequency found for COVID vaccination the clinical conclusion drawn from the results might better be the SPT positivity is not very useful. This should be better discussed.
RESPONSE: We now specified in the discussion that the clinical implications of our findings and the relevance of positive allergometric skin tests to PEG and PS80 is not yet clear, and more studies are necessary to assess whether patients with positive allergometric tests should actually be considered ineligible for vaccination (rows 206-209). Moreover, we emphasized that there might be other immunological mechanisms as well as other vaccine excipients involved in anti-SARS-CoV-2 vaccine hypersensitivity, thus more research is necessary to draw definitive conclusions (rows 260-263).
- A similar report has just been published by Montera et al. "The role of skin tests with polyethylene glycol and polysorbate 80 in the vaccination campaign for COVID-19: results from an Italian multicenter survey." (doi: 10.23822/EurAnnACI.1764-1489.291.). The results here should be critically compared to those in this report. In doing so it should be noted that Montera et al. have trouble with their arithmetic in that 10/362 is 2% not 0.02% and most people would not consider that rare.
RESPONSE: We discussed the results obtained by Montera et al. (specifying the correct % of positive allergometric tests reported) (rows 200-206).
- Given the high frequency of positive results the testing methodology and controls and what is considered positive becomes very important. The wide-ranging review of reference 15 does not give the information required. More precise methods are needed.
RESPONSE: We clarified our methods, including more details on patient selection criteria (rows 111-117) and test methodology (rows 123-133).
Round 2
Reviewer 1 Report
The manuscript has been revised with details and the limits of the study has been discussed objectively. I agree to publish this article.
Reviewer 3 Report
Satisfactory corrections have been made